# Early Detection of Male-Predominant Phenotypes in the Pattern of Ultrasonic Vocalizations Emitted by Autism Spectrum Disorder Model (*Crmp4*-Knockout) Mice

**DOI:** 10.3390/brainsci12050666

**Published:** 2022-05-20

**Authors:** Suzuka Shiono, Atsuhiro Tsutiya, Ritsuko Ohtani-Kaneko

**Affiliations:** 1Graduate School of Life Sciences, Toyo University, 1-1-1 Itakura, Oura 374-0193, Gunma, Japan; s39101900115@gmail.com; 2Clinical Proteomics and Molecular Medicine, St. Marianna University Graduate School of Medicine, 2-16-1 Sugao, Miyamae, Kawasaki 216-8511, Kanagawa, Japan; atsuchi@marianna-u.ac.jp

**Keywords:** autism spectrum disorder, collapsin response mediator protein 4, ultrasonic vocalizations, sex difference, model mouse

## Abstract

Male predominance is a known feature of autism spectrum disorder (ASD). Although ASD mouse models can be useful for elucidating mechanisms underlying abnormal behaviors relevant to human ASD, suitable models to analyze sex differences in ASD pathogenesis remain insufficient. Herein, we used collapsin response mediator protein 4 (*Crmp4*)-knockout (KO) mice exhibiting ASD-like phenotypes in a male-predominant manner and analyzed ultrasonic vocalizations (USVs) to detect potential differences between genotypes and sexes during the early postnatal period. We recorded isolation-induced USVs emitted from wild-type (WT) and *Crmp4*-KO littermates and compared the total number of USVs between genotypes and sexes. We classified USVs into 10 types based on internal pitch changes, lengths, and shapes and compared the number of USVs in each type by genotypes and sex. Male *Crmp4*-KO mice exhibited a reduction in the total number of USVs. *Crmp4*-KO decreased the number of USVs in 7 out of 10 USV types, and male KO mice exhibited a greater reduction than females in 3 of the 7 types. This study offers a suitable ASD animal model and tool for assessing sex-based communication deficits during the early postnatal period, both of which would be valuable for elucidating the underlying mechanism.

## 1. Introduction

Autism spectrum disorder (ASD) is a complex neurodevelopmental disorder. According to the Diagnostic and Statistical Manual of Mental Disorders, fifth edition (DSM-5), published by the American Psychiatric Association in 2013, ASD is characterized by persistent impairments in social communication and social interaction and restricted and/or repetitive behavioral patterns [1]. Notably, the prevalence rates of children with autism or those diagnosed with ASD have risen dramatically. The Centers for Disease Control and Prevention’s (CDC’s) Autism and Developmental Disabilities Monitoring Network and the World Health Organization (WHO) estimated that approximately 1 in 2000 individuals was diagnosed with autism in the 1970s, while the global average was recently estimated to be 1 in 100–160 children [2,3,4]. In addition, male predominance is a universally observed characteristic of ASD (the most frequently reported male–female ratio is approximately 4:1) [5,6]. The increasing number of children with ASD, as well as the observed sexual dimorphism, warrant the development of suitable animal models and analytical tools to approach underlying etiologies.

Mouse models of ASD act as useful tools for elucidating the mechanisms of abnormal behaviors relevant to the diagnostic symptoms of ASD in humans. To date, various ASD mouse models exhibiting ASD-like behaviors have been developed [7,8,9,10,11]. Using these model mice, altered physiological states such as abnormalities in excitatory/inhibitory balance [12,13] and/or altered neuronal morphology, such as developmental abnormalities in dendritic arborization, spine formation, and synaptic pruning [14,15,16,17], have provided evidence suggesting their involvement in the pathogenesis of ASD. In contrast, mechanisms for male predominance in ASD remain poorly understood, although “the female protective effect against ASD” and “the effect of perinatal androgen exposure in males” have been hypothesized [5,6,18,19]. Most studies in ASD model animals have been performed using males, partly due to the male predominance itself and partly to avoid the influence of the female estrus cycle, complicating behavioral evaluation. This could explain why the mechanism underlying sex differences in ASD remains unclear. Although recent studies focusing on sex differences in animal models of ASD have gradually increased [19,20,21], suitable models to analyze sex differences in ASD pathogenesis remain insufficient.

We have previously performed proteomic analysis and identified collapsin response mediator protein 4 (*Crmp4*) as a possible protein candidate mediating sexual differences in a certain sexual dimorphic nucleus in the preoptic area, i.e., the anteroventral periventricular nucleus (AVPV), during sexual differentiation [22]. Furthermore, whole-exome sequencing performed in the Nationwide Children’s Hospital and Department of Pediatrics (Ohio State University) has identified a de novo variant (S541Y) of *CRMP4* in a male patient with ASD [17]. In addition, we have shown that after *Crmp4*-knockout (KO) hippocampal neurons were transfected with *Crmp4*^S540Y^, which is homologous to human *CRMP4*^S541Y^, they had a significantly greater number of dendritic branching points than *Crmp4*-KO neurons transfected with wild-type (WT) *Crmp4* [17]. Neurons from *Crmp4*-KO mice (*Crmp4*-KO neurons) had significantly longer dendrites with more branching points than WT mice (WT neurons) [17]. These studies indicate that *Crmp4* deficiency or *Crmp4* missense mutations found in patients with ASD could alter dendritic morphology. Furthermore, *Crmp4*-KO mice exhibit sensory disorders, including olfaction and abnormalities in neuronal excitation in the olfactory bulb induced by a single odorant [23]. Moreover, we have reported that *Crmp4*-KO mice exhibit ASD-like behaviors and abnormalities in gene expression in a sex-dependent manner [8,17,23,24]. These studies indicate that *Crmp4*-KO mice provide a suitable animal model to examine mechanisms for some ASD-like phenotypes and their male predominance. 

Typically, standard social behavioral tests for mouse models of ASD, such as the social interaction test and three-chamber test, are performed using adolescent mice around or after puberty (≥6 w old). In contrast, analysis of ultrasonic vocalizations (USVs) can be performed using postnatal day (PD) 6–9 mouse pups. Accordingly, USVs emitted by ASD model pups are considered a useful tool for assessing social communication relevant to communication deficits, one of the core diagnostic criteria of ASD [25,26]. However, it has not been well studied whether USVs analysis can work as a suitable tool for assessing sex-based communication deficits during early postnatal periods of ASD mouse models.

In the present study, we used *Crmp4*-KO mice exhibiting some ASD-like phenotypes in a male-predominant manner and determined whether analyzing USVs can detect genotype- and sex-based differences in neonatal ASD model mice.

## 2. Materials and Methods

### 2.1. Animals

*Crmp4*-KO mice (Acc. No. CDB0637K: https://large.riken.jp/distribution/mutant-list.html, accessed on 16 May 2022) were established as previously described [27]. WT and *Crmp4*-KO mice were generated by mating *Crmp4*^+/−^ heterozygotes backcrossed to a C57BL/6N background. Mice underwent genotyping by polymerase chain reaction (PCR), as previously reported [27] with minor modifications. PCR amplification was performed for 35 cycles of 94 °C for 30 s, 54 °C for 15 s, and 72 °C for 1 min. All mice were housed in groups of 2–4 per cage, segregated by sex, in a room maintained a 12 h/12 h light-dark cycle and at 23 ± 1 °C. Food and water were provided ad libitum. Each male mouse was placed in a clean cage with one female mouse for mating. Pregnant mice were individually placed in cages. The use and care of animals were reviewed and approved by the Institutional Animal Care and Use and Ethical Committee of Toyo University.

### 2.2. USVs

USVs were detected according to previously reported methods [23,28]. USVs were detected using a condenser microphone designed to capture frequencies between 10 and 200 kHz (UltraSoundGate CM16; Avisoft Bioacoustics, Glienicke, Germany). The microphone was connected to an A/D converter (UltraSoundGate 116Hn; Avisoft Bioacoustics) at a sampling rate of 375 kHz, and acoustic signals were transmitted to a sound analysis system (SASLab Pro; Avisoft Bioacoustics).

Male and female *Crmp4*^+/−^ heterozygotes were mated and their offspring were used as USVs in response to maternal separation, as previously described [17,23]. Briefly, the mother and offspring were acclimatized to the behavior-testing room for 1 h. A pup was isolated in a clean glass beaker containing clean bedding at room temperature, and USVs were recorded for 5 min. After recording, pups were returned to their home cages and their genotypes were examined. Although the number of pups unexpectedly varied among four groups (male WT: n = 13, female WT: n = 26, male *Crmp4*-KO: n = 7, female *Crmp4*-KO: n = 9), we used all recorded data from them.

Spectrograms were generated with a fast Fourier transform length of 256 points and a time window overlap of 50% (100% frame size, Hamming window) to analyze USVs. A lower cut-off frequency of 20 kHz was used to reduce the background noise outside the relevant frequency band. We classified USVs into 10 types, including “Short”, “Flat”, “Upward”, “Downward”, “Modulated”, “Complex”, “One frequency jump”, “Frequency jumps”, “Mixed”, and “Chevron”, based on internal pitch changes, lengths, and shapes, based on previous criteria [29] with a slight modification according to Scattoni et al. [30].

### 2.3. Statistical Analysis

Data were analyzed using JMP version 9.0.2 (SAS Institute, Cary, NC, USA). We performed a two-way (genotype × sex) analysis of variance (ANOVA) followed by Tukey’s post-hoc test. For all analyses, *p* < 0.05 was considered statistically significant. 

## 3. Results

### 3.1. Total Number of USVs Was Decreased in Male Crmp4-KO Pups

We determined whether analyzing USV patterns can detect genotype- and sex-based phenotypic differences. Mice at PD7 were used for USV experiments, given that previous studies have shown that C57BL/6N mice produce a greater number of USVs at PD7 than at other postnatal stages [31,32]. Each pup was isolated from the nest in a clean glass beaker containing unfamiliar clean bedding at room temperature, and USVs were then recorded for 5 min (Figure 1).

The total number of isolation-induced USVs was compared to detect genotype- and sex-based differences (Figure 2). Two-way ANOVA revealed a significant difference between WT and *Crmp4*-KO mice (*p* < 0.001), with no differences noted between male and female pups (*p* = 0.216). Although the difference in total calls between female WT and female *Crmp4*-KO was not significant, this difference was significant between male WT and male *Crmp4*-KO mice, as determined using two-way ANOVA followed by Tukey’s post-hoc test (Figure 2). These findings indicated that *Crmp4* deficiency greatly decreased the total number of isolation-induced USVs in male pups than in female pups.

### 3.2. Crmp4-KO Decreased Specific USV Types in a Sex-Dependent Manner

Next, using data from recorded USVs (Figure 3a), including call duration and changes in internal pitch between maximum and minimum peak frequencies, we categorized USVs into 10 types according to previously reported criteria [29,30]. Table 1 and Figure 3b present the classification criteria and representative phenotypes, respectively.

The number of USVs in each type was compared between WT and *Crmp4*-KO mice and between sexes using two-way ANOVA (Figure 3c). Compared with WT pups, the numbers of USVs categorized as “Short”, “Flat”, “Downward”, “Modulated”, “One frequency jump”, “Frequency jump”, and “Mixed” were significantly reduced in *Crmp4*-KO pups (two-way ANOVA, “Short”, *p* = 0.049; “Flat”, *p* = 0.013; “Downward”, *p* < 0.001; “Modulated”, *p* = 0.009; “One frequency jump”, *p* = 0.021; “Frequency jump”, *p* = 0.047; and “Mixed”, *p* = 0.036). Conversely, the number of USVs categorized as “Upward”, “Complex”, and “Chevron” did not significantly differ between genotypes. Accordingly, these results indicated that *Crmp4* deficiency could significantly decrease the number of isolation-induced USVs in 7 of 10 USV types.

Among the 10 types of USVs, none significantly differed between sexes (Figure 3c). We observed no significant interaction between the two factors (genotype × sex) in the number of USVs categorized into 10 types, as determined by a two-way ANOVA. However, we found male-predominant effects of *Crmp4* deficiency in some types of USVs. Among above 7 types with significant differences between genotypes, the numbers of USVs categorized into 3 types, “Flat”, “Downward”, and “One frequency jump”, was significantly reduced in male *Crmp4*-KO pups when compared with male WT pups (*p* < 0.05, two-way ANOVA followed by Tukey’s post-hoc tests). On the other hand, no significant differences were observed in the numbers of USVs in these 3 categories between female WT and female *Crmp4*-KO pups. Therefore, male *Crmp4*-KO mice exhibited a greater reduction of USVs than females in 3 of the 7 types when compared to WT mice of the same sex. In addition, we observed a similar reduction tendency in the USVs of the type “Complex”, although the difference was not statistically significant.

## 4. Discussion

Isolation-induced USVs emitted by pups have been regarded as communicative signals of mother–pup interactions and are commonly studied to determine communication deficits in rodent models [25,32,33,34]. To establish a suitable animal model and analytical methods for elucidating sex differences in ASD pathogenesis before puberty, we used *Crmp4*-KO mice as an ASD model exhibiting impaired social behaviors in a sex-dependent manner [8,17,23]. Herein, our findings revealed that the total number of USVs (Figure 2) was significantly reduced in specific USV categories in male *Crmp4*-KO pups when compared with male and female WTs and female *Crmp4*-KO pups (Figure 3), indicating the suitability of *Crmp4*-KO pups as an animal model and analytical tool for elucidating male-predominance in ASD pathogenesis during the neonatal period.

Studies analyzing USVs have suggested that changes in the total number of isolation-induced USVs in various genetic ASD models are variable. In other words, the total number of USVs emitted from ASD model mice can be greater or less than those emitted by WTs. Caruso et al. (2020) have summarized the results of USV call rates among various ASD mouse models [35]. Compared with WT/control subjects, the call rate was decreased in *Avprf1bR*-, *Cd157*-, *Cadm1*-, *Cntnap2*-, *Dab1*-, *Fmr1*-, *Foxp1*-, *Foxp2*- *Gabra5*-, *Nlgn2*-, *Nrxn1**α*-, *NS-Pten*-, *Orpm*-, *Oxt*-, *Shank1*-, *Shank2*-, *Thp2*-KO pups, and *Foxp2*-mutated and *Nlgn3*-mutated pups, whereas it was increased in BTBR, *Mecp2*-KO, *Reln*^+/−^, *Sert-Ala56*-KO, and *Tsc1*-KO pups. In addition, the total number of isolation-induced USVs was decreased in a valproic acid-induced mouse model of autism and maternal immune activation mouse models, whereas it was increased in postnatal immune activation mouse models. 

Furthermore, USVs emitted from rodents are known to form a sequential structure consisting of syllables [36]. Mice and rats have been known to produce USVs as part of courtship behaviors [37,38], and USVs emitted from rats during courtship behavior have two-syllable categories, which are considered ‘pleasant’ and ‘distress’. Conversely, pups reportedly emit USVs, which play an important role in mother–pup communication [39,40], and can be classified into syllable categories according to their spectrotemporal patterns [29,30]. Caruso et al. (2020) reported the additional value of USV spectrographic analysis in ASD mouse models [35]. The total number of USVs, as well as the number of calls in each category, was used to assess the communication ability of ASD model pups.

Scattoni et al. (2008) have reported that BTBR mice, a known ASD mouse model, emit a significantly greater number of isolation-induced USVs than WT mice [32]. In addition, the number of calls belonging to three types, “Harmonics”, “Composite”, and “Two-syllable”, which are referred to as “Mixed”, “Modulated”, and “One frequency jump” in the present study, were significantly increased in ASD model pups when compared with WT mice. In addition, *Tbx1* heterozygous, an ASD-like phenotype, reportedly produce a smaller number of isolation-induced USVs categorized as “Complex”, “Two-syllable/One frequency jump”, “Frequency jump”, and “Flat” than those from WT pups [41]. In *NS-Pten* KO pups, the number of calls categorized as “Complex”, “Two-syllable/One frequency jump”, “Chevron”, and “Composite/Modulated” were found to be reduced, and those categorized as “Upward” and “Frequency step” were elevated [42]. In the present study, we showed that the number of calls categorized as “Two-syllable/One frequency jump”, “Flat”, and “Downward” was decreased in *Crmp4*-KO pups, when compared with those in WT mice. Combined with these results, these four ASD model pups commonly altered the number of USVs categorized into “Two-syllable/One frequency jump”.

It should be noted that only one study has examined sex-based differences in the number of USVs in each call-type category between WT and ASD model pups [43]. Fragile X syndrome is the most common inherited form of intellectual disability and a known monogenic cause of ASD. Fragile X syndrome is caused by a mutation in *FMR1*, which encodes a fragile X mental retardation protein. *Fmr1*-KO mice reportedly exhibit ASD symptoms and communication deficits. Nolan et al. (2020) have found no difference in the number of vocalizations among genotypes. However, male *Fmr1*-KO pups produced fewer “Complex”, “Composite/Modulated”, “Short”, “Downward”, and “Two-syllable/One frequency jump” calls, as well as a greater number of “Chevron” and “Frequency steps” than male WTs. Female *Fmr1*-KO pups reportedly produce fewer “Chevron”, “Short”, “Harmonics”, and “Downward” calls and a greater number of “Frequency steps” types than female WTs. Although studies on sex-related variations in USV types remain limited, our present results and those reported by Nolan et al. (2020) showed that the decrease in the number of “Two-syllable/One frequency jump” calls was commonly observed only in male ASD model pups, compared with their numbers in WTs [43]. Although functional meanings of “Two-syllable/One frequency jump” and other calls remain uncertain, these calls whose numbers were altered in ASD model pups may have implications for mother–pup communication. 

Behavioral tests for analyzing social communication, such as the three-chamber test and social interaction test, are generally performed using mice aged ≥ 6 w at the onset of puberty. In contrast, approximately 1-w postnatal pups can be used to analyze USVs. In addition, we found that, even at an early immature stage, male predominance can be observed in USVs emitted from *Crmp4*-KO ASD model mice. In conclusion, the present study provides a suitable ASD animal model and a tool for assessing male-predominant communication deficits during the early postnatal period, both of which will be useful for elucidating the underlying mechanism. 

## Figures and Tables

**Figure 1 brainsci-12-00666-f001:**
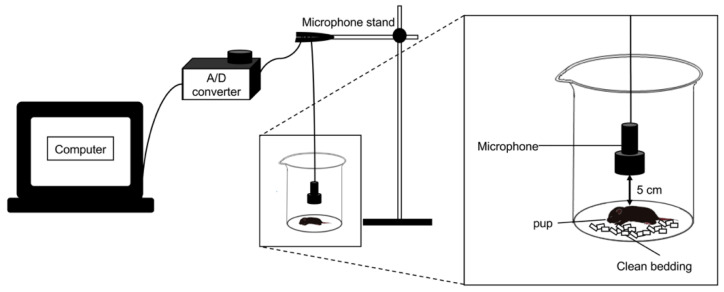
Experimental setting of ultrasonic vocalizations (USVs) recording. A condenser microphone was connected to an A/D converter and placed approximately 5 cm above a pup. The pup was isolated in a clean glass beaker containing clean bedding at room temperature for recording USVs.

**Figure 2 brainsci-12-00666-f002:**
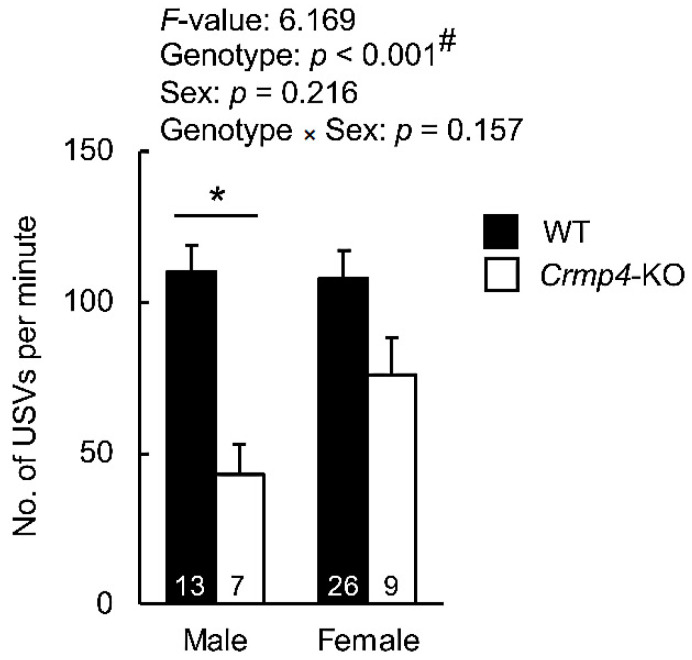
Total number of ultrasonic vocalizations (USVs) emitted from female and male wild-type (WT) and *Crmp4*-knockout (KO) pups for 5 min. Results from two-way analysis of variance (ANOVA) are shown in the top. Numbers in the bars indicate sample size. #, a significant difference at *p* < 0.05 (two-way ANOVA). *, a significant difference at *p* < 0.05 assessed by Tukey’s post-hoc test for multiple comparisons. Data are expressed as the mean ± standard error of the mean.

**Figure 3 brainsci-12-00666-f003:**
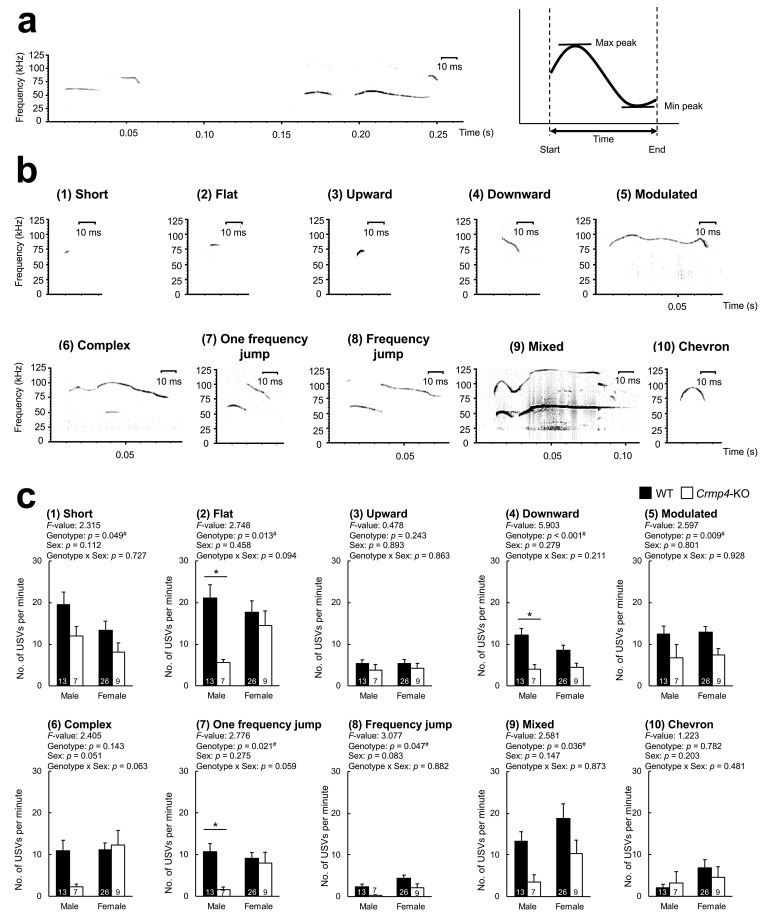
Representative sound spectrograms of ultrasonic vocalizations (USVs). (**a**) Sound spectrograms of representative USVs. An illustration of a spectrogram in the right part of (**a**) indicates maximum peak frequency, minimum peak frequency, and time (duration) of a call. (**b**) Representative sound spectrograms of 10 types of USVs. (**c**) The number of USVs in each of ten categories emitted from male and female wild-type (WT) and *Crmp4*-KO pups for 5 min. Results from two-way analysis of variance (ANOVA) are shown in the top. Numbers in the bars indicate sample size. #, a significant difference at *p* < 0.05 (two-way ANOVA). *, a significant differences at *p* < 0.05 assessed by Tukey’s post-hoc test for multiple comparisons. Data are expressed as the mean ± standard error of the mean.

**Table 1 brainsci-12-00666-t001:** Classification criteria of USVs.

Categories	Criteria
Short	Time < 5 ms and frequency modulation < 6.25 kHz.
Flat	Time modulation > 5 ms, and frequency modulation is <6.25 kHz.
Upward	Frequency increases in one direction, and modulation is ≥6.25 kHz.
Downward	The frequency decreases in one direction, and modulation is ≥6.25 kHz.
Modulated	The frequency is modulated in two or more directions, and the modulation is ≥6.25 kHz.
Complex	One or more additional frequency components (overtone or non-linear phenomenon, but not concentrated) are added, and the frequency range is not fixed.
One frequency jump	There is no time difference between continuous frequencies, one jump in which the frequency changes instantaneously, and no noise.
Frequency jumps	There is no time difference between continuous frequencies, multiple jumps in which the frequency changes instantaneously, and no noise.
Mixed	A frequency that has one or more jumps and is noisy.
Chevron	The frequency decreases by ≥6.25 kHz and increases by ≥12.5 kHz, and is similar to the “inverse U type”.

USVs, ultrasonic vocalizations.

## Data Availability

The datasets generated and/or analyzed during the current study are available from the corresponding author upon reasonable request.

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
