# Peer review of "Early Detection of Male-Predominant Phenotypes in the Pattern of Ultrasonic Vocalizations Emitted by Autism Spectrum Disorder Model (Crmp4-Knockout) Mice"

_brainsci, 2022, doi:10.3390/brainsci12050666_

Round 1
Reviewer 1 Report
The manuscript is well written and designed. The authors have already several publications with this model. There is a clear reduction in USV production in the mouse model used. The authors should the minor issues below.
The general quality of figures is low for publication; the authors should consider improve resolution of figures.
In methods authors mention: WT (male: n=13, female: n = 26) and Crmp4-KO (male: n = 7, female: n = 9). Is there any reason why a disparate number of animals in different groups?
For clarity purposes the authors should present the individual values/animals in figures.
In figure 2, the authors say “Different letters above bars indicate statistical differences “. But what exactly the letters “a”, “b”, and “ab” mean?
In results 3.2 change “gender-dependent” for “sex-dependent”
In table 1 the criteria for flat is ≤6.25kHz, and then criteria for downward or upward is ≥6.25kHz. Consider change flat for < or the others for >, as the value 6.25kHz is included in only one category.
The authors may consider some type of multiple comparisons correction especially for figure 3.
Author Response
Response to Reviewer 1 Comments
We appreciate the kind comments and helpful remarks from Reviewer 1. According to your remarks, we carefully revised our manuscript, table, figures, and figure legends as shown in the attached file.

Reviewer 2 Report
In the present manuscript, the authors report genotype- and sex-based alterations in the total number and profile of ultrasonic vocalizations (USVs) emitted by the collapsin response mediator protein 4 - knockout (Crmp4-KO) mice. In detail, they describe 1)the reduction in the number of USVs emitted by Crmp4-KO vs wildtype (WT) mice in 7 of 10 identified USV types, and 2)the reduction in the number of USVs emitted by males vs females KO mice in 3 of 10 USVs types.
There are some critical issues to be considered.
- Authors need to revise the similarity in the main text.
- The conclusions reached cannot be justified based on the results. The data here describe a significant effect of genotype in the pattern of USVs emitted by KO pups (similarly to other numerous ASD mouse models described in literature; Caruso et al., 2020) but does not show significant sex-linked differences that support the conclusions (and the title) of the manuscript.
- The question behind the research is poorly formulated. Indeed, the introduction provides a detailed description about the Crmp4-KO mouse model but fails to clearly outline the purpose of the present research.
- Regarding the research design, it is not clear the choice of the sample size (WT, n= 13 male and 26 female; Crmp4-KO, n= 7 male and 9 female).
- There are flaws in the presentation of the data. 1) In the graphs, the letters "a" and "b" above the bars are confusing and make it difficult to interpret the statistical differences. 2) In the figure 3C, the graphs have different scales (0 to 30; 0 to 20; 0 to 10): this does not allow for proper interpretation of the data.
- The authors describe a significant reduction in the numbers of USVs categorized as “flat”, “downwards” and “one frequency jump” in male Crmp4-KO pups when compared with male WT pups. Because there are no differences observed between female WT and female Crmp4-KO pups, the authors conclude that Crmp4-KO deficiency reduces the numbers of isolation -induced USVs in male pups than in females in 3 of the 10 types identified. However, looking at the graphs in Figure 3C, the differences described for the "downward" type are not apparent. On the contrary, it would seem there are similar differences to those described by the authors for the type "complex" that however is not taken in consideration.
Author Response
Response to Reviewer 2 Comments
We appreciate the helpful comments and suggestions from Reviewer 2. According to your remarks, we carefully revised our manuscript, figures, and figure legends as shown in the attached file.

Reviewer 3 Report
Review 1: This study aimed to analyze the potential differences between genotypes and sexes during the early postnatal period using collapsin response mediator protein 4 (Crmp4)-knockout (KO) mice in ultrasonic vocalizations test. They found that Male Crmp4-KO mice exhibited a reduction in the total number of USVs. Crmp4-KO decreased the number of USVs in 7 out of 10 USV types, and male KO mice exhibited a greater reduction than females in 3 of the 7 types. The finding offers a suitable ASD animal model and tool for assessing sex-based communication deficits during the early postnatal period, both of which would be valuable for elucidating the underlying mechanism.
Strengths:
The experimental design is completed and rigorous, and the data is presented clearly. The statistical analysis is good and test methods were chosen correctly. The USVs testing method is very good and detailed.
Weakness:
- The F values of the statistical results should be provided.
- Do the Crmp4-related signal pathways show any sex differences in wild-type or ko mice?
Author Response
Response to Reviewer 3 Comments
We appreciate the kind advice and question from Reviewer 3. According to your advice, we revised our figures as shown in the attached file.

Round 2
Reviewer 2 Report
I am satisfied with the authors' responses and I recommend publication in the journal.